# Training Universal Adversarial Perturbations with Alternating Loss Functions

**Deniz Sen, Berat Tuna Karli, Alptekin Temizel**

Graduate School of Informatics, Middle East Technical University
deniz.sen_01@metu.edu.tr, tuna.karli@metu.edu.tr, atemizel@metu.edu.tr

## Abstract

Despite being very successful, deep learning models were shown to be vulnerable to crafted perturbations. Furthermore, changing the prediction of a network over any image by learning a single universal adversarial perturbation (UAP) was shown to be possible. In this work, we propose 3 different ways of training UAPs that can attain a predefined fooling rate, while, in association, optimizing $L_2$ or $L_\infty$ norms. To stabilize around a predefined fooling rate, we have integrated an alternating loss function scheme that changes the current loss function based on a given condition. In particular, the loss functions we propose are Batch Alternating Loss, Epoch-Batch Alternating Loss, and Progressive Alternating Loss. In addition, we empirically observed that UAPs that are learned by minimization attacks contain strong image-like features around the edges; hence we propose integrating a circular masking operation to the training to alleviate visible perturbations further. The proposed $L_2$ Progressive Alternating Loss method outperforms the widespread attacks by providing a higher fooling rate at equal $L_2$ norms. Furthermore, Filtered Progressive Alternating Loss can further reduce the $L_2$ norm by 33.3% at the same fooling rate. When optimized with regards to $L_\infty$, Progressive Alternating Loss manages to stabilize on the desired fooling rate of 95% with only 1 percentage point of deviation, despite $L_\infty$ norm being particularly sensitive to small updates.

## Introduction

Due to their success, deep learning models have been adopted as standard methods in many visual tasks. On the other hand, deep neural networks have also been shown to be vulnerable against purposefully generated data samples called adversarial examples. The most popular way of generating adversarial examples is applying an adversarial attack to a benign sample and obtaining a particular perturbation that leads to misclassification when added to this sample. With this method, generating a whole dataset of adversarial examples involve putting each image through the same algorithm to calculate an image-dependent perturbation, which results in a significant time overhead. Recently, it has been shown that a single perturbation can be used to make any sample an adversarial example; these perturbations are called universal adversarial perturbations (UAP).

Table 1: Overview of the proposed UAP training methods

| Attack | Abbreviation | Loss Alteration Condition |
|---|---|---|
| Batch Alternating Loss | B-AL | Fooling rate of each batch |
| Epoch-Batch Alternating Loss | EB-AL | Fooling rate of each batch, if the previous epoch reached the fooling rate |
| Progressive Alternating Loss | P-AL | Fooling rate up to the point of processing the current batch |
| Filtered Progressive Alternating Loss | FP-AL | Same as P-AL but after each batch the filter in Equation 3 is applied on the UAP |

These types of perturbations have distinct properties compared to image-dependent adversarial perturbations, such as having image-like features by themselves (Zhang et al. 2020), whereas traditional perturbations are perceived as noise by humans. Another point where image-dependent attacks differ from UAPs is that it is straightforward to fool a predefined ratio of a target dataset; applying the attacking algorithm only to several samples can achieve this, as most image-dependent attacks can achieve fooling rates close to 100%. However, since UAPs tend to form image-like features from only a small set of data, it is likely to force misprediction in the rest of the data. Therefore, some regulation is required in the training procedure of the UAPs to achieve the target fooling rate exactly.

In this work, we show that UAPs can be trained using alternating loss strategy, which switches the loss function (between adversarial and $L_p$ norm losses) based on the fooling rate performance of the current state of the UAP. As the alternating loss scheme is not directly applicable to UAP training, we propose 3 algorithms (Table 1) adapting this scheme to UAP training. Furthermore, we show that UAP features naturally accumulate around the edges of the perturbation vectors; using this information, we apply a circular mask to

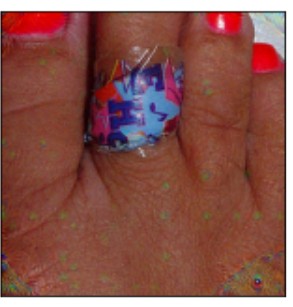 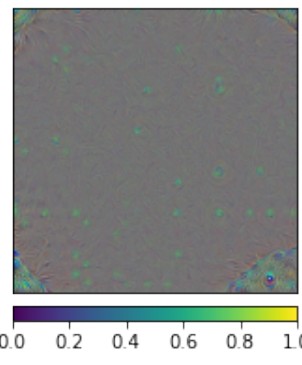 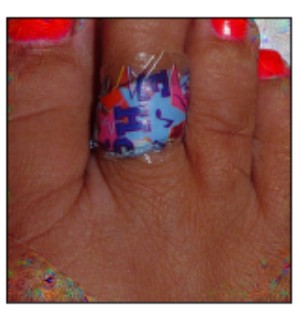 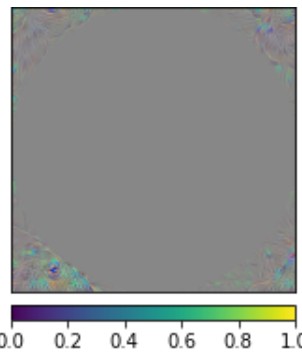

Figure 1: Sample UAPs trained with $L_2$ P-AL (left) and Filtered P-AL (right). The prediction for benign image is the correct class, *band aid*, with 98.70% confidence. The adversarial examples yield *peacock* predictions with 99.90% and 99.97% confidence respectively. The predictions are from ResNet50.

the UAPs during training to eliminate visible distortions on the actual target object.

## Related Work

Adversarial perturbations are traditionally generated specifically for a single sample. Fast gradient sign method (FGSM) (Goodfellow, Shlens, and Szegedy 2014) is an adversarial attack that can be used with $L_1$, $L_2$ and $L_\infty$ norms, and despite its simplicity, it is still being widely used. Basic iterative method (Kurakin, Goodfellow, and Bengio 2017), and projected gradient descent (Madry et al. 2018) algorithms; as opposed to FGSM, iteratively optimize the perturbation with fixed-size steps. Different from these $L_p$ bounded attacks, there are also minimization attacks. DeepFool (Moosavi-Dezfooli, Fawzi, and Frossard 2016) aims to geometrically shift the benign image to the closest decision boundary to force misclassification. Carlini&Wagner attack (Carlini and Wagner 2017) reformulates a constrained optimization problem to generate the smallest successful adversarial perturbation. Perceptual Color distance Alternating Loss (Zhao, Liu, and Larson 2020) is a modified version of Carlini&Wagner that decouples the norm and adversarial optimization using the alternating loss method, which is also adopted in our proposed algorithms.

Universal adversarial perturbations were formally introduced in (Moosavi-Dezfooli et al. 2017), which applies DeepFool (Moosavi-Dezfooli, Fawzi, and Frossard 2016) algorithm to each sample iteratively, updates the overall universal perturbation, and projects the perturbation to a $L_p$ ball. Generative models were also trained to obtain UAPs. Network for adversary generation (NAG) (Mopuri et al. 2018) is a generative adversarial network framework that trains a generator, using a frozen target classification network, to generate a UAP, from an input noise vector. On the other hand, Fast Feature Fool (Mopuri, Garg, and Babu 2017) is a data-free algorithm that trains a UAP that maximizes the activation values of convolutional layers. This algorithm generally performs worse than data-dependent attacks but is good proof that UAPs can be generated by only using the properties of the target convolutional network.

Feature-UAP (Zhang et al. 2020) is a $L_p$ constrained attack that trains a UAP using mini-batch training to achieve state-of-the-art fooling rates, and the authors provide a detailed comparison between image-dependent attacks and universal attacks. High-Pass-UAP (Zhang et al. 2021) is a similar algorithm that also trains UAPs using mini-batches but also applies a Fourier domain high-pass filter to the current UAP, after revealing that UAPs tend to perform better when they contain more high-frequency features while being imperceptible to the human eye. In the same work, Universal Secret Adversarial Perturbation (Zhang et al. 2021) was introduced, where a UAP not only fools models but also contains extractable information. Training UAPs to make a network to perceive a predefined class as another target class was introduced and named 'Double Targeted UAPs' (Benz et al. 2020).

In this paper, we propose 3 alternative approaches using alternating loss for training UAPs: Batch Alternating Loss (B-AL), Epoch-Batch Alternating Loss (EB-AL), and Progressive Alternating Loss Training (P-AL). All the universal attacks in the literature are norm bounded; thus, norm optimization is not a stochastic operation; instead, it is a projection. Our method is different from these works in this regard. In addition, we propose integrating filtering to the training further to reduce the perturbations at the same fooling levels.

## Methodology

The universal adversarial attack problem can be formally defined as in Equation 1, where $v$ is the UAP, $x$ is a benign image sampled from a dataset $\mu$, $f$ is the target model, $\delta$ is the minimum fooling rate, and $\epsilon$ is the maximum $L_p$ norm of $v$.

$$P_{x\sim\mu}(f(x + v) \neq f(x)) \geq \delta \text{ s.t.} \quad \|\nu\|_p \leq \epsilon \quad (1)$$

The norm bounded attack concept is widely used in adversarial machine learning; however, it is also possible to formulate it as a minimization problem as in Equation 2, by slight modifications over Equation 1.

Algorithm 1: Batch Alternating Loss Training(B-AL)
___

**Input**: Dataset $\mu$, target class $t$, target fooling rate $\delta$, epoch $k$, model $f$, norm $p$
**Variables**: Counter $i$, fooling rate $fr$, prediction $out$, adversarial loss function $adv$, loss $L$
**Output**: Universal adversarial perturbation $v$

  1: $v \leftarrow 0$
  2: $i \leftarrow 0$
  3: **while** $i < k$ **do**
  4:     **for** $x \sim \mu$ **do**
  5:         $out \leftarrow f(x + v)$
  6:         $fr \leftarrow$ # of incorrect predictions / batch size
  7:         **if** $fr < \delta$ **then**
  8:             $L \leftarrow adv(out, t)$
  9:         **else**
 10:             $L \leftarrow ||v||_p$
 11:         **end if**
 12:         backpropagate $L$
 13:         update $v$
 14:         $i \leftarrow i + 1$
 15:     **end for**
 16: **end while**
 17: **return** $v$
___

$$\text{minimize} \, ||v||_p \quad \text{s.t.} \quad P_{x \sim \mu}(f(x + v) = t) \approx \delta \qquad (2)$$

The variable $t$ is the target class, and $\delta$ is the target fooling rate in this equation. Now, the problem becomes finding the smallest $||v||_p$, which attains the desired fooling rate. This problem can be turned into a standard minimization based UAP problem by setting $\delta$ to 1, as then the constraint will simply become $f(x + v) = t$. Different from the problem defined in 1, this problem cannot be solved by only adding a regularizer (such as clamping the UAP into $\epsilon$ norm) to the training, as the probability constraint requires the whole dataset to be a part of the regularization. For that, training a UAP first to fool a large portion of the dataset, then gradually diminishing the norm of the UAP while bringing the fooling rate down to the desired level is a suitable way. This theoretical framework can be implemented using alternating loss functions.

In this work, we introduce 3 UAP training methods (shown in Table 1), leading to different attacks that take advantage of the alternating loss strategy. Alternating loss scheme switches between 2 loss functions depending on the current state of the training; this strategy is used in image dependent adversarial attacks (optimize the norm of the perturbation if the current image is adversarial, if not, optimize the adversarial loss); however, it is not directly applicable to the UAP domain. The first proposed method is Batch Alternating Loss (B-AL), aiming to reach the desired fooling rate by achieving the same fooling rate for each batch. The second method is Epoch-Batch Alternating Loss (EB-AL), which considers the fooling rate achieved over the epoch, alongside each batch. The final training method is Progressive Alternating Loss (P-AL), which uses the fooling rate

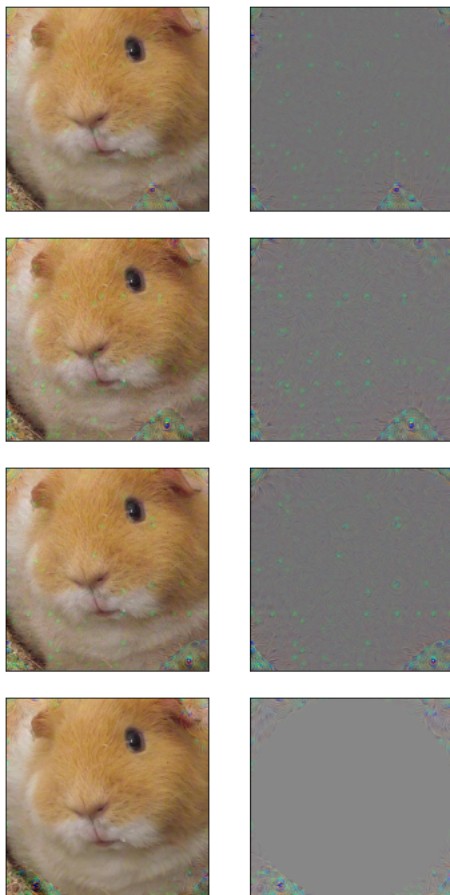

Figure 2: UAP calculated for *Peacock* target class applied over a hamster image (left column) and the UAP images (right column), corresponding to B-AL, EB-AL, P-AL, Filtered P-AL, from top to bottom.

achieved until the current batch to alter the loss function. We also empirically find that stronger features are generated around the edges, along with smaller artifacts in the middle; therefore, we propose applying filtering during training to alleviate these artifacts. The proposed filtering scheme can be integrated into any minimization-based UAP training.

The alternating loss scheme requires a decoupled decision mechanism that will set the current loss function to either adversarial loss, which changes the prediction of the network, or the norm of the UAP, which is either $L_2$ or $L_\infty$. In image-dependent attacks, alternating loss function can be iteratively selected based on the current state of the perturbation; when the current perturbation is successful in making the image an adversarial example, minimize the norm of the perturbation, else, optimize the adversarial loss to obtain an adversarial example (Zhao, Liu, and Larson 2020). However, training a UAP for several iterations for a single image while changing the loss function at each iteration would be incompatible with mini-batch training. Instead, we can use batches to train the UAP while changing the current loss

**Algorithm 2**: Epoch-Batch Alternating Loss Training (EB-AL)

**Input**: Dataset $\mu$, target class $t$, fooling rate $\delta$, epoch $k$, model $f$, norm $p$
**Variables**: Counter $i$, fooling rate $fr$, prediction $out$, adversarial loss function $adv$, loss $L$, optimization mode $m$, number of correct predictions $correct$, image number counter $imcount$, fooling rate over the epoch $epochfr$
**Output**: Universal adversarial perturbation $v$

```
 1:  v ← 0
 2:  i ← 0
 3:  m ← 'epoch'
 4:  while i < k do
 5:      correct ← 0
 6:      imcount ← 0
 7:      for x ∼ μ do
 8:          out ← f(x + v)
 9:          correct ← correct+ # of correct predictions
10:          imcount ← imcount+ batch size
11:          fr ← # of incorrect predictions / batch size
12:          if (m == 'epoch') or
                 (m == 'batch' and fr < δ) then
13:              L ← adv(out, t)
14:          else
15:              L ← ||v||_p
16:          end if
17:          backpropagate L
18:          update v
19:          i ← i + 1
20:      end for
21:      epochfr ← 1 − correct/imcount
22:      if epochfr < δ then
23:          m ← 'epoch'
24:      else
25:          m ← 'batch'
26:      end if
27:  end while
28:  return v
```

**Algorithm 3**: Filtered Progressive Alternating Loss Training (FP-AL)

**Input**: Dataset $\mu$, target class $t$, fooling rate $\delta$, epoch $k$, model $f$, norm $p$, mask radius $D$
**Variables**: Counter $i$, fooling rate $fr$, prediction $out$, adversarial loss function $adv$, loss $L$, number of correct predictions $correct$, image number counter $imcount$, circlar filter $filter$
**Output**: Universal adversarial perturbation v

```
 1:  v ← 0
 2:  i ← 0
 3:  filter ← filter in Equation 3 with D
 4:  while i < k do
 5:      correct ← 0
 6:      imcount ← 0
 7:      for x ∼ μ do
 8:          out ← f(x + v)
 9:          correct ← correct + # of correct predictions
10:          imcount ← imcount + batch size
11:          fr ← (imcount− length of correct) / imcount
12:          if fr < δ then
13:              L ← adv(out, t)
14:          else
15:              L ← ||v||_p
16:          end if
17:          backpropagate L
18:          update v
19:          v ← filter(v)
20:          i ← i + 1
21:      end for
22:  end while
23:  return v
```

To address this problem, the following 2 methods of training are proposed.

### Epoch-Batch Alternating Loss (EB-AL)

Algorithm 2 shows the pseudo-code of EB-AL. This method ensures that the UAP does not start diminishing the adversarial energy until the desired fooling rate is achieved. Before reaching the target, the loss function strictly becomes adversarial, regardless of the individual performance of each batch. At the end of each epoch, we check whether the target fooling rate was achieved; if it was, then the same loss function alteration scheme presented in B-AL training is applied in the next epoch. EB-AL almost always achieves the desired fooling rate, if it is possible at all. When an epoch is completed with a successful fooling rate, many batches may yield fooling rates above the target. This causes the extensive usage of the norm loss function, which brings the overall fooling rate down, making the loss function strictly adversarial in the next epoch. This phenomenon makes the fooling rate oscillate around the target, which may cause some imprecision in attaining the target fool rate.

function based on the fooling rate achieved on that batch. Note that the main parameter in this optimization is the desired fooling rate over the whole training dataset.

### Batch Alternating Loss (B-AL)

Algorithm 1 shows the pseudo-code of B-AL. In this approach, the loss function is switched according to the fooling rate performance of the current state of the UAP over the current batch: if UAP can achieve the desired fooling rate over the current batch, norm loss is selected; otherwise, adversarial loss is selected, which is chosen to be the cross-entropy function. This training method can bring the fooling rate around the desired level in several epochs. However, when some batches yield the desired fooling rate, the loss function lowers the adversarial energy to decrease the norm of the UAP; thus, the overall fooling rate stays below the target fooling rate since most of the images remain benign.

Table 2: $L_2$ attack results, provided in terms of $L_2$ and $L_\infty$ metrics and FR refers to the Fooling Rate. Note that UAP (Moosavi-Dezfooli et al. 2017) and F-UAP attacks (Zhang et al. 2020) are set to reach the same $L_2$ values as P-AL to allow comparison of FR at the same level of perturbation. The proposed attacks that cannot reach 95% fooling rate are italicized, and are not considered while selecting the best results.

| Method | DenseNet121 | | | ResNet50 | | | GoogleNet | | | VGG16 | | |
|--------|------|------------|------|------|------------|------|------|------------|------|------|------------|------|
|        | L2   | $L_\infty$ | FR   | L2   | $L_\infty$ | FR   | L2   | $L_\infty$ | FR   | L2   | $L_\infty$ | FR   |
| B-AL   | *9.14*  | *0.45* | *0.93* | *9.08*  | *0.52* | *0.93* | *9.56*  | *0.45* | *0.91* | 7.10 | 0.47 | **0.95** |
| EB-AL  | 14.11 | 0.47 | **0.98** | 14.36 | 0.60 | **0.98** | 15.73 | 0.58 | **0.98** | 7.39 | 0.44 | **0.95** |
| P-AL   | **11.16** | 0.43 | 0.95 | **11.66** | 0.53 | 0.95 | **13.01** | 0.60 | 0.95 | **5.52** | 0.44 | **0.95** |
| UAP    | 11.16 | 0.29 | 0.33 | 11.66 | **0.21** | 0.34 | 13.01 | **0.27** | 0.43 | 5.52 | **0.15** | 0.30 |
| F-UAP  | 11.16 | **0.23** | 0.90 | 11.66 | 0.32 | 0.93 | 13.01 | 0.34 | 0.91 | 5.52 | 0.16 | 0.70 |

Table 3: $L_\infty$ attack results, provided in terms of $L_2$ and $L_\infty$ metrics and FR refers to the Fooling Rate. Note that UAP (Moosavi-Dezfooli et al. 2017) and F-UAP attacks (Zhang et al. 2020) are set to reach the same $L_\infty$ values as P-AL to allow comparison of FR at the same level of perturbation.

| Method | DenseNet121 | | | ResNet50 | | | GoogleNet | | | VGG16 | | |
|--------|------------|------|------|------------|------|------|------------|------|------|------------|------|------|
|        | $L_\infty$ | $L_2$ | FR   | $L_\infty$ | $L_2$ | FR   | $L_\infty$ | $L_2$ | FR   | $L_\infty$ | $L_2$ | FR   |
| B-AL   | 0.17 | 24.96 | **1.00** | 0.17 | 26.64 | **1.00** | 0.20 | 26.63 | **1.00** | 0.16 | 19.04 | **1.00** |
| EB-AL  | 0.16 | 24.42 | **1.00** | 0.18 | 29.39 | **1.00** | 0.22 | 33.55 | **1.00** | 0.17 | 25.70 | **1.00** |
| P-AL   | **0.11** | **16.76** | 0.96 | **0.13** | **16.61** | 0.96 | **0.16** | **20.60** | 0.96 | **0.11** | **12.45** | 0.95 |
| UAP    | 0.11 | 25.71 | 0.52 | 0.13 | 29.64 | 0.60 | 0.16 | 35.32 | 0.79 | 0.11 | 25.84 | 0.75 |
| F-UAP  | 0.11 | 25.55 | 0.99 | 0.13 | 26.74 | 0.99 | 0.16 | 31.00 | 0.99 | 0.11 | 24.57 | 0.99 |

## Progressive Alternating Loss (P-AL)

Algorithm 3 shows the pseudo-code of P-AL. Because of the nature of the training procedure, it is not trivial to have the fooling rate completely stabilized over the target; however, it is possible to minimize the oscillation caused by the phenomena explained in the previous section. In P-AL training, similar to EB-AL, the adversarial loss is maintained until the target fooling rate is achieved. Furthermore, after reaching the target, the loss function is altered based on the fooling rate achieved from the beginning of the epoch until the currently optimized batch. This way, it is possible to maintain the overall fooling rate while optimizing the norm when possible. Although this method cannot completely prevent the oscillation, it minimizes it to a certain degree.

## Masked Training

We empirically found out that when the UAP is not normalized by a norm constraint, perturbations with high intensities accumulate around the edges and corners. These perturbations also contain more image-like features; thus, they influence the prediction towards the target. Applying masks to smooth out perturbations was investigated for image-dependent attacks (Aksoy and Temizel 2020), and it is a simple yet efficient way to control the geometry of the perturbations; hence we propose integrating a masking operation in the UAP training procedure. To reduce the perturbations around the center of the image (which are also mostly positioned on top of the target object), we apply a filter (Equation 3) to the UAP after each batch, where $(x, y)$ is the pixel position, $h$ and $w$ are the height-width values of UAP respectively, and $D$ is the radius of the circle. Algorithm 3 shows FP-AL training, which is the P-AL training with filtering.

The visual effect of filtered training can be seen in Figure 1.

$$f(x) = \begin{cases} 1, & \text{if } \sqrt{(\frac{w}{2} - x)^2 + (\frac{h}{2} - y)^2} \geq D \\ 0, & \text{otherwise} \end{cases} \quad (3)$$

This method can be applied to any minimization-based UAP training scenario, as for the norm constrained attacks, the perturbations do not always mitigate towards the edges. Also, we empirically found that smooth circular filters such as 2D Gaussian or Butterworth (Butterworth et al. 1930) filters tend to limit the adversarial capacity of the UAPs, by slightly smoothing the features around the edges.

## Experimental Design

We have trained the UAPs using a sampled ImageNet dataset containing 10000 images, formed by 10 images from each class. We have compared our attack with 2 other attacks which can be trained with small dataset sizes; vanilla UAP (Moosavi-Dezfooli et al. 2017), and Feature-UAP attacks (Zhang et al. 2020). We have chosen the target class as *peacock* for our attacks and Feature-UAP (vanilla UAP is strictly an untargeted attack as it is based on DeepFool). Similar to our attack, vanilla UAP allows a target fooling rate specification over the training set. Therefore to allow comparisons on the same ground, we set this parameter to be the same, specifically, a target fooling rate of 95%. However, as both of these attacks are norm constrained attacks, a direct comparison is not possible; therefore, we first trained UAPs with our attacks, then we set the constraints, i.e., epsilons to match our obtained $L_2$ or $L_\infty$ values depending on the type of the norm which was selected to be optimized. To measure the performance of the attacks, we have used the

Table 4: Comparison between the results of P-AL and FP-AL, in both $L_2$ and $L_\infty$. FR signifies the fooling rate over the whole dataset.

| Method | DenseNet121 | | | ResNet50 | | | GoogleNet | | | VGG16 | | |
|---|---|---|---|---|---|---|---|---|---|---|---|---|
| | $L_2$ | $L_\infty$ | FR | $L_2$ | $L_\infty$ | FR | $L_2$ | $L_\infty$ | FR | $L_2$ | $L_\infty$ | FR |
| $L_2$ P-AL | 11.16 | **0.43** | **0.95** | 11.66 | **0.53** | **0.95** | 13.01 | 0.60 | **0.95** | 5.52 | 0.44 | **0.95** |
| $L_2$ FP-AL | **8.38** | **0.41** | **0.95** | **10.75** | 0.60 | **0.95** | **10.62** | **0.52** | **0.95** | **8.93** | **0.43** | **0.95** |
| $L_\infty$ P-AL | 16.76 | **0.11** | **0.96** | 16.61 | **0.13** | **0.96** | 20.60 | **0.16** | **0.96** | **12.45** | **0.11** | **0.95** |
| $L_\infty$ FP-AL | **15.57** | 0.15 | 0.97 | **16.42** | 0.15 | **0.96** | **18.80** | 0.18 | 0.97 | 13.28 | 0.13 | 0.96 |

standard ImageNet validation set having 50000 images. For fast convergence, Adam was selected as the optimizer, and the UAPs have been trained for 20 epochs. For the experiments where filtering is applied, a radius of 112 is used, as the dimensions of the input images are $224 \times 224$.

## Results

Two different experiments have been conducted by applying the attacks with $L_2$ and $L_\infty$ norms, which are supported by all attack types in question. The results are then compared with regards to both $L_2$ and $L_\infty$ values.

### $L_2$ attacks

Table 2 shows the $L_2$ attack results for B-AL, EB-AL, P-AL, vanilla UAP and Feature-UAP (F-UAP). For 95% fooling rate constrained attacks, the attack is regarded successful if it is at or above the target. Despite achieving the smallest $L_2$ value compared to the other base models, B-AL cannot attain the target FR for DenseNet121 (Huang et al. 2018), ResNet50 (He et al. 2015) and GoogleNet (Szegedy et al. 2014). It can only reach the target for VGG16 (Simonyan and Zisserman 2015) with batch normalization; however, in that case, its $L_2$ results are comparatively higher. On the other hand, EB-AL is above the target FR by 3 percentage points (except for VGG16 where it achieves the target FR), which renders the attack sub-optimal; furthermore, the $L_2$ values are consistently higher than both B-AL and P-AL. P-AL, which was introduced to address the inefficiencies of B-AL and EB-AL, consistently achieves the desired fooling rate while also having the smallest $L_2$ values. Overall, P-AL stays inside the desired range; furthermore, by only integrating a filter during training (Table 4), FP-AL achieves even lower perturbation levels, both in terms of $L_2$ and distance from the desired fooling rate. This algorithm consistently achieves a 95% fooling rate while yielding the smallest - successful- $L_2$ distance over any given attacks.

It should be noted that both UAP and F-UAP are meant to be mainly run under $L_\infty$ constraints; however, the algorithms are suitable for $L_2$ normalization during training. As mentioned earlier, both of these attacks are norm-constrained as opposed to our minimization attacks which makes it difficult to compare them. However, when the $L_2$ constraints are equalized at the level of our attacks, we see that both UAP and F-UAP fall below the desired fooling rate; nonetheless, both reach significantly smaller $L_\infty$ values compared to our attacks.

### $L_\infty$ attacks

Table 3 shows the results of the $L_\infty$ attacks. It should be noted that while our attacks are mainly designed to minimize $L_2$ norms of the UAPs, they can minimize $L_\infty$ as well. This time, B-AL and EB-AL overshoot the desired fooling rate, which is not optimal in our constraints; besides, their $L_\infty$ values are comparatively higher. P-AL achieves better $L_2$ and $L_\infty$ values while staying closer to the desired fooling rate. According to the results in Table 4, FP-AL slightly increases the $L_\infty$ values while also getting further from the target, in exchange for an overall decrease in $L_2$ norm.

As UAP and F-UAP are mainly $L_\infty$ bounded attacks, we can expect better results from them. UAP shows much better results than $L_2$ normalized attack. However, still cannot reach the target fooling rate against either of the networks. F-UAP achieves a 99% fooling rate for each network type, albeit yielding comparatively higher $L_2$ values.

## Discussion

Altering the loss function based on the progressive fooling rate gives the best results in both $L_2$ and $L_\infty$ attacks. Although our other attacks take a similar approach, because each batch affects the optimization too much, the alternating loss scheme causes unstable behavior, thus making it harder to converge to the desired fooling rate. Another possible drawback from these attacks is that the batch size plays a crucial role in how the optimization proceeds. For instance, the final UAPs trained with batch sizes of 32 and 128 may have severe performance differences since by increasing the sample size, we get a better understanding of the performance over the whole dataset. P-AL is independent of the batch size, and it is more stable around the desired fooling rate.

We should also point out that P-AL addresses the problem of obtaining a fooling rate around the target level, not exceeding it to obtain even a better fooling rate. If the target is to maximize the fooling rate as much as possible, our attacks will likely be less optimal. For instance, if we set the target fooling rate as 100%, the UAP will be trained only with the adversarial loss function, therefore never minimizing the $L_p$ norm. EB-AL may be the better choice to maximize the fooling rate since it will first bring the fooling rate to 100% if possible; then, it will try to minimize the norm. Table 2 and 3 shows that higher fooling rates can be achieved by EB-AL, although the $L_p$ norms are slightly higher. On that note, using the $L_\infty$ norm to optimize the fooling rate can be another way of maximizing the fooling rate, along

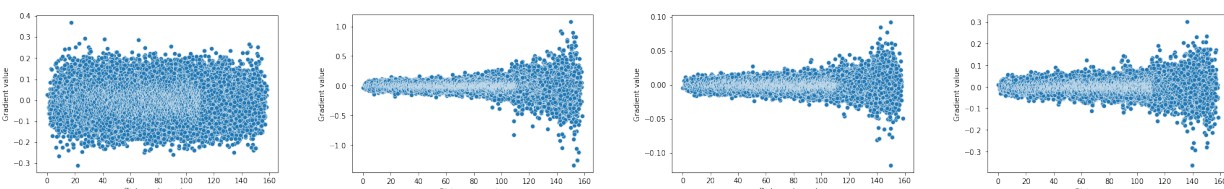

Figure 3: Vertical axes show the mean gradient value, horizontal axes show the distance between the pixel containing the corresponding mean gradient, and the center of the image. The scatter plots are extracted from UAP states after iteration number 1, 30, 150 and 300.

with B-AL and EB-AL. Using $L_\infty$ usually makes the optimization converge faster at high fooling rate targets; since $L_\infty$ norm is in s scale [0-1], the adversarial loss function (cross-entropy) takes much higher values, thus updates the perturbation values more drastically. In those cases, while optimizing the UAP in a strong adversarial manner, a small norm optimization is also done, yielding a UAP with a very high fooling rate.

### Perturbation Features

Figure 2 shows sample UAPs trained with different methods. The UAPs on the first 3 rows (obtained without a filter) exhibit perturbations with visible image feature accumulated around the edges. However, this phenomenon is not caused by the alternating loss; rather, it is a consequence of performing targeted universal attacks that minimize the target standard loss. The gradients on the perturbation are high over the edges after only a few iterations, which causes the image features to concentrate on these regions. Figure 3 shows the scatter plots of the mean gradient values when a UAP is applied on the whole dataset versus the distance of each pixel containing the gradient. The scatter plots are generated using a state of a trained UAP; from left to right, top to bottom, the UAP is taken from iterations 1, 30, 150 and 300 of the training. The gradients flow to the pixels far from the center of the images. We speculate that, since usually the main objects of the images are located around the center, the magnitudes of gradients with respect to a loss function whose objective class is different from the original class become relatively higher where features from the original object are absent, hence the edges and corners. On the other hand, it is also possible to see small feature-full perturbations that are generated around the center of the image, such as the green dots that can be seen in Figure 2. It is known that universal perturbations take advantage of image features that outweigh the original image features; hence, it can be possible to understand the target class by only looking at the universal perturbations; yet, the small accumulations around the center defy these assumptions. For that reason, our filtered training scheme not only makes the center of attention clean of perturbations but also quantitatively yields better $L_p$ norms and stable fooling rates.

### Conclusion

This work proposes and evaluates alternative approaches for training a UAP that can achieve target fooling rates for a dataset while being a minimization optimization, rather than being $L_p$ bounded. We have integrated 'alternating loss,' an image-dependent attack strategy, into the universal adversarial domain. As it was not directly possible to incorporate this strategy into a training procedure, we came up with 3 different approaches for its utilization. B-AL training altered the loss function based solely on the currently processed batch. EB-AL training also took the performance of the UAP over the whole dataset before changing the loss function. Finally, P-AL training considered the fooling rate up to the point where a batch is processed. We achieved remarkable $L_2$ distances using P-AL while maintaining the desired fooling rates. On top of P-AL, we have also applied circular filtering to mitigate the small perturbations that appear in the center of the UAP to the edges. In this way, we obtain perceptually better UAPs, by not having perturbations in the center of the image while achieving even smaller $L_2$ distances. On the other hand, this work can further be improved by regularizing the altered loss functions to achieve better $L_p$ norms. Also, investigating the mitigation of the image-features on the UAPs can also be helpful for understanding not only the existence of these perturbations, but also the behaviour of the deep neural networks.

## Acknowledgements

This work has been funded by The Scientific and Technological Research Council of Turkey, ARDEB 1001 Research Projects Programme project no: 120E093

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
