# OpenReview forum: "Training Universal Adversarial Perturbations with Alternating Loss Functions"
_AAAI.org/2022/Workshop/AdvML — AAAI-22 AdvML Workshop LongPaper_

### Official Review · Reviewer_DeJ9 · 2021-11-29
**Methods using alternating loss functions to generate UAP**

**Rating:** 6
**Confidence:** 4

**Review:**

This paper proposed to train UAP with alternating loss functions, including three types of schedule. Further, a circle mask is adopted to reduce the $\ell_2$ and $\ell_\infty$ norm of the perturbations with the same fooling rate. The experiments are conducted on a subset of ImageNet and the results show the proposed methods, especially P-AL, get closer to the desired fooling rate.

Strengths:
* The combination of alternating loss function scheme and the training of UAP is interesting.
* The analysis of perturbations distribution supports the empirical claim that perturbations accumulate around the edges.

Weaknesses:
* The writing has some mistakes. For instance, the fooling rate in algorithms is not defined well, even contradicts itself in algorithm 2. Also, grammar mistakes are common.
* The motivation of this alternating scheme is not clear.
* The formulation of UAP training might be mathematically wrong, especially Equation 2. $P\approx \delta$ should be a constraint, not a term of $\min$. Also, I doubt the claim that '' this problem can be turned into a min-max problem as well by setting $\delta$ to 1''.
* Some statements are confusing. The bold numbers in the tables seem not to be the greatest or the smallest. Also, for the metric of fooling rate, I don't see why it's proper to say that the closer to the desired fooling rate the better the method is.
* More analysis of the filter mask (e.g., shape, diameter) can be provided.

---

### Official Review · Reviewer_ZC8i · 2021-11-30
**Alternating Loss Functions for UAP**

**Rating:** 7
**Confidence:** 4

**Review:**

This paper proposes a new method based on the idea to optimize the norm of adversarial examples. The alternating loss function involves 2 branches: optimize norm or optimize the attack loss. This setting works extremely well in generating unviersal adversarial perturbation. The proposed method outperform the baseline method supported by the experiments on Imagenet.

To improve the paper, I think the structure could be optimized by claiming the contributions in the introduction. In all, it is a good paper with clear idea and massive experiments.

---

### Decision · Program_Chairs · 2021-12-01

**Decision:**

Accept (Long Paper)

**Comment:**

Both reviewers agree to accept this paper. Please address the comments of reviewers in the final version.